# PRP of T2DM Patient Immobilized on PCL Nanofibers Stimulate Endothelial Cells Proliferation

**DOI:** 10.3390/ijms24098262

**Published:** 2023-05-05

**Authors:** Anastasiya O. Solovieva, Natalia A. Sitnikova, Vadim V. Nimaev, Elena A. Koroleva, Anton M. Manakhov

**Affiliations:** Research Institute of Clinical and Experimental Lymphology—Branch of the ICG SB RAS, 2 Timakova Str., 630060 Novosibirsk, Russia

**Keywords:** Type 2 Diabetes Mellitus (T2DM), diabetic foot ulcers (DFU), polycaprolactone (PCL) nanofibers, platelet-rich plasma (PRP) therapy

## Abstract

Diabetic foot ulcers (DFU) are a common complication of Type 2 Diabetes Mellitus (T2DM). Development of bioactive wound healing covers is an important task in medicine. The use of autologous platelet-rich plasma (PRP) consisting of growth factors, cytokines and components of extracellular matrix is a perspective approach for DFU treatment, but we previously found that some T2DM PRP samples have a toxic effect on mesenchymal stem cells (MSCs) in vitro. Here, we covalently immobilized T2DM PRP proteins on polycaprolactone (PCL) nanofibers, and the growth of endothelial cells on the PCL-COOH-PRP was investigated. Additionally, the level of NO reflecting the cytotoxic effects of PRP, angiogenin, and VEGF levels were measured in T2DM PRP samples. The results showed that the application of PCL-COOH-PRP nanofibers allows to remove the cytotoxicity of T2DM PRP and to improve endothelial cell adhesion and proliferative activity. We showed that the origin of T2DM PRP (the level of PRP toxicity or presence/absence of DFU) does not influence the efficiency of cell growth on PCL-COOH-PRP, and on the level of angiogenin, vascular epidermal growth factor (VEGF) in PRP itself.

## 1. Introduction

The prevalence of Type 2 Diabetes Mellitus (T2DM) steadily increased in both developing and developed countries and according to various sources, it increased from 371 million in 2013 to 463 million in 2019 worldwide [1]. Chronic wound injury or diabetic foot ulcers (DFU) is a frequent complication of type 2 diabetes and represents an economic and social burden [2]. To date, standard treatment protocols do not lead to a complete solution to this problem, resulting in a high incidence of amputations and even mortality in this category of patients. Tissue reconstruction in this pathology is one of the urgent tasks in biomedicine. There are some therapies that demonstrate good efficacy, e.g., the treatment of wounds with hyperbaric oxygen and negative pressure [3,4], the use of skin substitutes as wound dressings [5,6,7,8], the therapy with autologous platelet-rich plasma (PRP) [9,10,11,12,13,14], and the combination of both last methods [15,16,17].

The skin substitutes are biopolymers with structures that mimic the extracellular matrix. They form a suitable substrate that mimics the mechanical, chemical, and structural properties of the tissue, as well as protecting the wound from bacterial invasion and providing gas exchange [5,6,7,8].

Autologous PRP was proposed for the therapy of chronic non-healing wounds, including DFU in T2DM patients [9,10,11,12,13,14]. Researchers are interested in it because PRP contains growth factors, cytokines, and components of the extracellular matrix that are very active. PRP helps stem cells grow and move by bringing macrophages to the damaged area. It also controls the cytokine background of the wound, reduces inflammation, and helps new blood vessels grow. However, the results of the use of PRP for the treatment of chronic wounds showed its efficiency in only 60% of cases [12]. Several factors can be a cause of ineffective PRP therapy in some T2DM patients with DFU. Firstly, in chronic wounds, the release of growth factors from PRP may be hindered by over-cleaving by cellular or bacterial proteases. Secondly, the toxicity of PRP is related to the formation of toxic metabolites of glucose, reactive lipid metabolites, and NO metabolites in the blood of T2DM patients [18,19,20]. Intracellular accumulation of toxic metabolites can trigger various pathological signaling pathways, including inflammation, production of reactive oxygen species (ROS), altered Ca^2+^ signaling, and mitochondrial dysfunction [21]. Nevertheless, the use of PRP for DFU therapy is actively studied, and methods for preserving the factors that make up its composition and prolong its action are in development [13,14].

It seems attractive to combine the advantages of biopolymers, which structurally and chemically mimic the tissue, with the therapeutic bioactive proteins of PRP to stimulate the mechanisms involved in tissue repair. It was demonstrated, by us [22,23,24] and some other groups [6,7], that the immobilization of PRP onto the matrix promotes cell adhesion, migration and proliferation, and, ultimately, accelerates wound healing.

In this study, polycaprolactone (PCL) nanofibers modified with COOH groups were used as a matrix for covalent immobilization of PRPs from T2DM patients. The effect of PRP-modified nanofibers on the viability and proliferative activity of endothelial cells, as well as the effect on their secretion of nitric oxide, was evaluated. In addition, the content of angiogenic factors (angiogenin, Vascular Endothelial Growth Factor (VEGF)) and NO concentration in toxic and nontoxic T2DM PRP samples were determined.

## 2. Results

### 2.1. Optimization of PCL Nanofibers Structure and PRP Immobilization

The morphology and chemical composition of nanofibers were thoroughly discussed in our previous study [24,25]. The PCL nanofibers were randomly oriented and had a diameter of ~400 nm. The subsequent deposition of the plasma polymer resulted in minor alterations to the nanofibers’ thickness but no major changes to their topography. On the contrary, the chemistry of the PCL-ref and PCL-COOH were extremely different. Water contact angle (WCA) experiments clearly demonstrated the effect of plasma coating. The PCL-ref displayed a WCA of 120 ± 2° (Figure 1a). The WCA was significantly reduced during the plasma layer deposition (down to 43 ± 3°), as shown in Figure 1b.

Polar group grafting was responsible for the enhanced wettability of plasma-coated PCL nanofibers. The XPS analysis revealed quantifiable differences in the surface chemistries. Table 1 displays the XPS-measured chemical compositions of PCL nanofibers (PCL-ref) and PCL-COOH samples. Using the high-resolution spectra of each component, the atomic percentages of the elements were determined.

A sum of three components can be used to fit the XPS C1s spectrum of PCL-ref, which includes hydrocarbons CH_x_ (BE = 285 eV), ether group C-O (BE = 286.4 eV), and ester group C(O)O (BE = 289.0 eV), as shown in Figure 2a. The full width at half maximum (FWHM) of C-O was adjusted to 1.35 eV, while the FWHM of CH_x_ and C(O)O were set to 1.1 and 0.95 eV, respectively. Using a sum of four components, PCL-COOH was also fitted. These components included hydrocarbons CH_x_ (BE = 285.0 eV, which were used for BE scale calibration), carbon adjacent to carboxylic acid or ester group C*-C(O)O (BE = 285.5 ± 0.05 eV), carbon singly bonded to oxygen C-O (BE = 286.55 ± 0.05 eV), carbon doubly bonded to oxygen C=O/O-C-O (BE = 288.0 ± 0.05 eV), and carbon of ester or carboxylic group C(O)O (BE = 289.2 ± 0.05 eV). The concentrations of all components can be found in Figure 2b.

After immobilization of PRP onto PCL-COOH, the WCA practically was not changed 41 ± 4° (Figure 1c). In contact, the XPS analysis revealed that the immobilization of PRP led to significant compositional changes. PCL-COOH-PRP did, in fact, have a significant nitrogen concentration of 11.8 at%. The increase in the nitrogen concentration was accompanied by a significant decrease in the oxygen concentration. Additionally, 0.3% sulfur was also detected.

The XPS C1 curve fitting of PCL-COOH-PRP was performed using the same methodology as for the PCL-COOH and depicted in Figure 2c. After PRP was immobilized, the concentration of the C(O)O environment decreased while a new contribution associated with amides (N–C=O, BE = 288.3 eV) emerged. This finding supports the idea that biomolecules with a protein nature were affixed to the surface. Additionally, the N1s spectra for both samples were identical, showing a single component attributed to amide bonds (N-C=O) at a BE of 399.9 eV (Figure 2d).

All these findings suggested that PRP was significantly incorporated into our layers. Our special curve fitting presented in previous work revealed that more than 50% of the surface is covered by PRP [26]. Hence, the surface of the nanofibers was successfully modified with the PRP proteins.

### 2.2. The Influence of Immobilization of Toxic PRP on PCL-COOH Nanofibers on MSCs Viability

The application of autologous PRP for the treatment of DFU in T2DM patients seems tempting. However, we previously demonstrated that the addition of 5% T2DM PRP to the culture medium of MSCs led to a decrease in cell viability by more than 40% in 39% of PRP samples [27]. According to the results the following groups were defined: PRPnt, nontoxic, PRPtox, toxic group, or PRPther, therapeutic (from patients without DFU), and PRPsur, surgery group (from patients with DFU). PRP from healthy donors (PRPhealthy) was used as a control group. A group of PRP patients with T2DM was designated as PRP T2DM. 

Some data [28] showed that cytotoxicity of T2DM plasmas are related to the high toxic metabolites concentration. Peroxinitrite is one of the metabolites that iNOS produces from NO under oxidative stress conditions. To overcome this obstacle, PRP proteins of T2DM patients were covalently immobilized on PCL nanofibers. Previously, we showed that the adhesion, proliferation, and differentiation of mesenchymal stromal cells (MSCs) and fibroblasts were higher on PCL-COOH-PRP (PRP from healthy donors) as compared to unmodified PCL nanofibers [22,23]. We proposed that the immobilization followed by washing will allow to decrease in the cytotoxicity of T2DM PRP. Here, we compared the toxic effects of PRPtox versus PRPhealthy samples, which were added to cultural medium or immobilized on PCL-COOH on MSCs.

The data showed that the cultivation of MSCs in medium with PRPtox lead to cell death by necrosis (Figure 3D). After the immobilization of PRPtox on PCL-COOH and following cultivation the MSCs on it, the nuclei morphology corresponded to morphology of viable cells (structure, shape, density, the presence of mitosis). The effect of PRP samples on cell viability was determined by counting all fields of view after cell staining with Hoechst 33342. The data in Figure 3G show a significant increase in cell numbers growing on PCL-COOH-PRPtox versus on cultural plates with PRPtox addition. Our data suggest that immobilization of PRPtox on PCL-COOH eliminated the toxic effects and, at the same time, maintained the proliferative activity of MSCs.

### 2.3. Influence of PCL-COOH-PRP of T2DM Patients on Proliferation and NO Secretion of Endothelial Cells

Since activation of angiogenesis is necessary for DFU healing, the endothelial cells were used as a model cell system. We studied the cytotoxicity of PRPtox and PRPnt samples using endothelial cell lines. The cytotoxicity was evaluated on cultural plates, also using PCL-COOH samples with immobilized PRP, and the influence of these samples on cell viability was investigated by MTT-test and by counting of cell number. It was shown that PRPtox led to a significant decrease in the proliferative and mitochondrial activity of endothelial cells without signs of cell death by necrosis (Figure 4A).

To determine if there was any observed difference in the remaining cytotoxicity of PRP after its immobilization on PCL, 10 PRP samples from each of the PRPnt, PRPtox, and PRP-healthy groups were randomly chosen, immobilized, and the endothelial cell growth investigated. PCL-COOH without PRP and cells seeded on grow on cultural plates treated with type IV collagen were used as a controls. The data on cell morphology (Figure 5A) showed that the cells efficiently adhered and proliferated on PCL-COOH-PRP nanofibers, independently of the PRP origin. The number of cells and their density on PCL-COOH were lower than those on PCL-COOH-PRP. It should be noted that, normally, endothelial cells grow on cultural plates treated with type IV collagen. The data on cell count (Figure 4A) showed that the number of cells growing on PCL-COOH-PRP was 2.5-fold higher than the value for PCL-COOH.

Additionally, the total concentration of NO secreted by endothelial cells after its cultivation on PCL-COOH-PRP nanofibers was investigated.

The measurement of NO concentration in cultural medium of endothelial cells showed that NO level correlated with the number of cells (primary data are not shown) and it was 20–30-fold lower than the level in corresponding PRP samples (Figure 5A–C). After normalization of NO concentration to the cell count (Figure 4B), the values became approximately the same for both PRP-modified PCL and PCL-COOH nanofibers. The results suggest that the endothelial cells growing on PCL-COOH-PRP nanofibers secrete NO at the background level and its contribution in cytotoxicity of T2DM PRP seemed to be negligible.

Taken together, our results demonstrate (Figure 3 and Figure 5) that the immobilization of T2DM PRP on PCL nanofibers allows to remove the cytotoxicity of PRP and has a positive influence on adhesion and proliferation of MSCs and endothelial cells as compared to control PCL-COOH nanofibers.

### 2.4. NO, VEGF and Angiogenin Levels in T2DM PRP Samples

We estimated the contribution of NO concentration to the toxicity of PRP samples. The data showed (Figure 5A,C) that the level of NO in T2DM PRP was reliably higher (*p* = 0.003) as compared to PRP of healthy donors, and the increase in NO concentration was determined by the contribution of PRP of toxic group in T2DM PRP group (PRPhealthy/PRPtox, *p* = 0.005). The comparison with PRP from the nontoxic group revealed no reliable differences (PRPhealthy/PRPnt, *p* = 0.13). As a result, the toxicity of T2DM PRP correlated with higher levels of NO in the samples. The data in Figure 3B demonstrate that the distribution of T2DM PRP by surgery and therapy groups was not eligible, as the medians of PRPther and PRPsur were close to each other and the 25–75% quartiles were significantly overlapped.

Next, the levels of growth factors stimulating angiogenesis, such as angiogenin and VEGF, were measured in PRP samples. The data showed (Figure 5D,G) that in T2DM PRP, the level of angiogenin increased (*p* = 0.001) and the level of VEGF decreased (*p* = 0.002) relative to PRP from healthy donors. The data depicted in Figure 5E,F,H,I revealed that there were no differences in the values between therapy and surgery or toxic and nontoxic groups. Thus, the levels of both growth factors under study as well as NO concentration do not depend on the presence of DFU in T2DM patients. In addition, the levels of angiogenin and VEGF do not depend on the toxicity of T2DM PRP. 

## 3. Discussion

One of the frequent complications of T2DM is diabetic foot syndrome that can lead to DFU. Treatment with suitable dressings is an important part of the management of DFUs and requires both optimization of blood sugar level and surgical interference [29]. According to the recommendations of guidance on DFU treatment, after surgically cleaning the wound, it is important to choose a suitable dressing [30]. Here, PCL nanofibers modified by PRP of T2DM patients as potential dressing for DFU were investigated. In the DFU area, a local bloodstream is destroyed. The endothelial cells are the main participants in blood perfusion recovery. To evaluate the potential of PCL-COOH-PRP as matrix for cell growth, the endothelial cells were used as a model cell system. Previously, we showed that the covalent immobilization of PRP proteins on PCL promote the adhesion and proliferation of MSCs [22] and fibroblasts [23]. The results showed [23] that the fibroblasts growing on PCL-COOH-PRP, when reached monolayer were oriented in the same direction. Such orientation of cells is, apparently, due to the contact inhibition of isotropic movement of the cells. In other words, the fibroblasts forming monolayer on PCL-COOH-PRP were arranged in tissue-like manner. Additionally, we tested PCL-COOH-PRP nanofibers as the wound dressing in mice with inherent T2DM [24]. The results showed that the wound healing was faster using PCL-COOH-PRP as compared to unmodified PCL. Taken together, our data confirm the high potential of PRP-modified PCL as wound dressing.

In patients with DFU, the efficiency of autologous PRP therapy was observed only in 60% of cases [12]. Our data are in accordance with this results. Thus, we previously showed [27] that 39% of PRP samples display cytotoxic effects in MSCs. In this work, we showed that PRPtox samples also have a strong inhibitory effect on proliferation of endothelial cells. Our data demonstrated (Figure 5A–C) that this toxicity correlates with a high NO level in the PRP samples. NO concentration increase due to hyperglycemia in blood of T2DM patients was shown by other scientist groups [31].

NO synthesis is carried out by the family of NO synthases (NOS), which are represented by three different isoforms: neuronal (nNOS), inducible (iNOS2), and endothelial (eNOS). The main source of NO is normally associated with eNOS activity in endothelial cells. NO is a short-lived, gaseous free radical that has a variety of biological effects in both physiological and pathological conditions. For endothelial cells, it is one of the key molecules that supports the survival of endothelial cells by inhibiting apoptosis caused by pro-inflammatory cytokines and pro-atherosclerotic factors, including reactive oxygen species [32,33]. Under pathological conditions such as inflammation and hyperglycemia, the inhibition of eNOS in endothelial cells and activation of iNOS in neutrophils and macrophages occurs [34]. The level of NO rises, while the level of reactive oxygen species (ROS) rises as well. ROS oxidize NO, forming peroxynitrites, which in turn oxidize the zinc thiolate center of endothelial NO synthases (eNOS) [35]. Dysfunctional eNOS leads to a decrease in the secretion of NO by endothelial cells and its biological activity [20]. According to our data, the NO concentration in PRP T2DM (Figure 5) was more than 10-fold higher than that secreted by endothelial cells (Figure 4B). Our data confirm that the main source of NO determined in plasma samples is not endothelial cells. The immobilization of PRP on PCL and following washing allow to remove all toxic metabolites from PRP (including NO, glucose and reactive lipid metabolites, which detected in the blood of T2DM patients [18,19,20]). Another possible source of NO is NO secretion by endothelial cells growing on PCL-COOH-PRP nanofibers. The results demonstrated (Figure 4B) that the endothelial cells seeded on PCL containing or not PRP secretes NO at background level independently of the PRP origin (toxic or nontoxic). Low NO level correlates to high survival and proliferative activity of endothelial cells growing on PCL-PRP.

Thus, the immobilization of PRP on PCL completely reduces the cytotoxicity and allows to use PRP from any T2DM patient.

Another criterion which can impact the efficiency of cell growth on PCL-COOH-PRP is the growth factor levels in PRP of different origins. Here, the levels of angiogenin and VEGF in PRP samples were measured. The data showed (Figure 5D,G) that in T2DM PRP, the increase in angiogenin (*p* = 0.001) and the decrease in VEGF level (*p* = 0.002) as compared to PRP of healthy donors was observed. In the literature, there are different data on the angiogenin and VEGF levels in PRP. Thus, in T2DM PRP, the VEGF level increased [36], and increased or decreased [37] versus plasma of healthy donors depending on the presence of mutations in defined positions of the gene. The systemic meta-analysis of data on the angiogenin level [38] revealed no statistically significant differences between blood plasma of T2DM and healthy donors. Our results on angiogenin and VEGF levels (Figure 5D–I) showed that the division of T2DM PRP into groups (nontoxic/toxic PRP or PRP of patients without/with DFU) is not eligible, and it reflects only the difference between the growth factors levels in T2DM PRP versus PRP of healthy donors. These data, taken together with data on decreased toxicity of T2DM PRP due to immobilization, suggest that the efficiency of autologous PRP-modified PCL does not depend on the presence of acute ulcer process and on the accumulation of toxic NO metabolites in blood of T2DM patients.

In conclusion, in this study, we used PCL nanofibers as carriers of autologous T2DM PRP. The covalent immobilization of PRP proteins allows to remove the toxic metabolites. We showed that the use of PCL-COOH-PRP nanofibers as matrix allows to enhance the adhesion and proliferation of endothelial cells independently on PRP origin (from T2DM or healthy donors).

## 4. Materials and Methods

### 4.1. Blood Plasma Samples of T2D Patients

The study involved 115 T2D patients: 60 patients without of DFU (therapeutic department of clinic NIICEL—Group PRPther) and 55 of patients with DFU (from surgery department of clinic NIICEL—Group PRPsur). Exclusion criteria were severe concomitant pathology, skin infectious diseases in the acute stage, acute inflammatory processes, critical limb ischemia, pregnancy and lactation, and malignant neoplasms. The control group consisted of 15 age-matched men and women who did not have a verified diagnosis of T2D and who did not smoke or drink alcohol for at least a week before blood samples taking (Group PRPhealthy). Blood platelet rich plasma samples were prepared as previously described [19]. Plasma was collected and stored at −70 °C until use. The study protocol was approved by the local ethics committee on 12 October 2017 (protocol no. 135). The informed consent of the patient to the examination was taken in accordance with the directives of the European Community (86/609/EEC) and the Helsinki Declaration, in compliance with the Ethical Principles for Scientific Medical Research with Human Participation and in accordance with the Rules of Clinical Practice in Russian Federation. 

### 4.2. ELISA Detection of Plasma Level of VEGF and Angiogenin

Concentration of VEGF and angiogenin in plasma samples were estimated by Human Angiogenin ELISA Kit (Sigma Aldrich, Darmstadt, Germany) and Human VEGF-A ELISA (eBioscience, San Diego, CA, USA) was used according to manufacturer’s recommendations. 

### 4.3. Plasma Nitric Oxide (NO) Measurement

Detection of total nitric oxide (nitrite + nitrate) was performed with Griess Reagent kit (Invitrogen, Waltham, MA, USA). In brief, blood plasma was deproteinized with 10% TCA (Trichloroacetic Acid) and the sample was centrifuged at 14,000 rpm for 20 min at 4 °C to remove the precipitated protein. Nitrates are reduced to nitrites by nitrate reductase (Sigma-Aldrich, St. Louis, MI, USA), The resulting supernatant was used to measure the NO levels according to manufacturer’s instructions. The absorbance of the solution was read on a plate reader at 540 nm. To quantify the NO production, a standard curve was generated using sodium nitrite.

### 4.4. Preparation of Nanofibers

Nanofibers were produced by electrospinning a 9-weight percent solution of PCL (80,000 g/mol) solution. The sample processing can be found elsewhere [39]. Briefly, acetic acid (99%) and formic acid (98%) were used to dissolve the granulated PCL. All substances were bought from Sigma Aldrich (Darmstadt, Germany). The acetic acid (AA) and formic acid (FA) had a weight ratio of 2:1. The samples were electrospun by a Super ES-2 machine produced by ESpin Nanotech, (ESpin Nanotech, Kanpur, India), which included both drum and static plate collectors. In this investigation, a static collector plate was used for collecting the nanofibers. The flowrate of PCL solution was 1 mL/h. The samples were collected onto polypropylene fabric and placed at 12 cm distance from the nozzle. The electrospinining voltage was kept at 50 kV. The authors used the term PCL-ref to refer to the untreated, as-prepared PCL nanofibers.

### 4.5. Plasma-Coated COOH

The plasma polymerization technique for Ar/CO_2_/C_2_H_4_ was discussed in full elsewhere [40,41]. On Si wafers and PCL nanofibers, COOH plasma polymer layers were deposited utilizing a UVN-2M vacuum system equipped with rotary and oil diffusion pumps. The reactor’s residual pressure was below 10^−3^ Pa. A radio frequency (RF) Cito 1310-ACNA-N37A-FF power supply (Comet, Flamatt, Switzerland) connected to an RFPG-128 disk generator (Beams and Plasmas, Moscow, Russia) located in the vacuum chamber was used to ignite the plasma. The duty cycle and RF power were, respectively, set to 5% and 500 W. In the vacuum chamber, CO_2_ (99.995%), Ar (99.998%), and C_2_H_4_ (99.95%) were introduced. Ar gas flow was fixed at 50 sccm, whereas CO_2_ and C_2_H_4_ gas fluxes were ste to 35 and 10 sccm, respectively. They were managed with the aid of a 647C Multi-Gas Controller (MKST, Newport, RI, USA). Using a VMB-14 unit (Tokamak Company, Dubna, Russia) and D395-90-000 BOC Edwards controllers, the chamber pressure was measured. Then, 8 cm was established as the distance between the RF electrode and the substrate. The time allotted for the deposition was 15 min. The plasma coated nanofibers were denoted as PCL-COOH.

### 4.6. Coating of Scaffolds with PRP

Prior to immobilization of PRP, all samples were sterilized under ultraviolet (UV) light for 45 min. The PCL-COOH was immersed in a 1-ethyl-3-(3-dimethylaminopropyl) carbodiimide (EDC) (Sigma Aldrich, St. Louis, MO, USA, 98%) solution in water (2 mg/mL) for 15 min. The sample was carefully washed thrice with PBS and was then incubated with studied samples of PRP for 15 min at room temperature. After the reaction, the sample was thoroughly washed with PBS. The samples were denoted as PCL-COOH-PRP. The samples of PRP of patients with T2DM were separated on groups: PCL-COOH-PRPtox (n = 12) and PCL-COOH-PRPnt (n = 12) in accordance with the data that were published by us earlier.

### 4.7. Chemical Characterization of Samples

The sample chemical characterization was performed by X-ray photoelectron spectroscopy (XPS). The XPS analysis was carried out using a PHI5500VersaProbeII instrument (Ulvac PHI, Osaka, Japan) equipped with a monochromatic Al Kα X-ray source (hν = 1486.6 eV) at a pass energy of 23.5 eV and X-ray power of 50 W. The spectra were fitted using the CasaXPS software version 2.3.25 (Casa Software Ltd., Teignmouth, UK) after subtracting the Shirley-type background. The maximum lateral resolution of the analyzed area was 0.7 mm. The binding energies (BEs) for all carbon and oxygen environments were taken from the literature [42,43,44,45].

An OCA 15 PRO device (Dataphysics, Filderstadt, Germany) outfitted with a measuring video system with a USB camera and a high-speed measuring lens with a movable viewing angle was used to measure the contact angles on the samples. The syringe’s needle had a 0.51 mm diameter. It was tested using distilled water. The drops had a one-liter (L) capacity. The sessile drop method was used to measure the values of the contact angles. According to the Young–Laplace algorithm, the calculated contact angle was determined. In 5–10 different areas of the substrate, droplets were generated for surface characterization. The values collected were averaged.

### 4.8. Cell Tests

Human endothelial cells were kindly provided by the Developmental Epigenetics Laboratory of Institute of Cytology and Genetics, Siberian Branch of Russian Academy of Sciences (Novosibirsk, Russian). Cells were cultured in medium (DMEM, Sigma Aldrich, St. Louis, MO, USA) supplemented with 10% fetal bovine serum (Gibco) on cultural plastics was coated by human collagen IV under standard culture conditions (humidified atmosphere, 5% CO_2_ and 95% air, at 37 °C). Then, cells were seeded on PCL-COOH-PRP and cultivated for 24 h. Control cells were cultivated on collagen-coated plastic dishes and on PCL-COOH nanofibers only. Then, culture medium was refreshed and added L-arginine for 30 min, and then, NO concentration in culture medium was assayed by Griess Reagent kit (Invitrogen).

Human mesenchymal stromal cells were extracted from bone marrow using standard methods and cultured in Dulbecco’s modified Eagle’s Medium: Nutrient Mixture F-12 (DMEM/F12, Sigma Aldrich, Paisley PA4 9RF, UK) that was supplemented with 10% fetal bovine serum (FBS, Gibco, Carlsbad, CA, USA). Cells were seeded in 96-well plates on scaffolds in concentration of 5 × 103 cells/well. The study was approved by the Ethics Committee of the RICEL-branch of ICG SBRAS (No 115 from 24 December 2015).

Cell viability were assayed by MTT colorimetric test. Cell count were detected by nuclei staining with Hoechst 33342. For 15 min at room temperature. Later, the cells were observed with a fluorescence microscope (Zeiss, Axio observer Z1, Oberkochen, Germany).

### 4.9. Statistical Analysis

For all results, the mean values and associated error standard deviations were calculated. For statistical analysis, we used Statistica 8 software. The *p* value ≤ 0.05 was considered statistically significant. The Shapiro–Wilk test was used to control the normal distribution of all variables. Since the obtained data did not show a normal distribution, a nonparametric test was appropriate, namely the non-parametric Mann–Whitney U test was used for comparing two groups and Kruskal–Wallis test was used for comparing more than two groups.

## Figures and Tables

**Figure 1 ijms-24-08262-f001:**
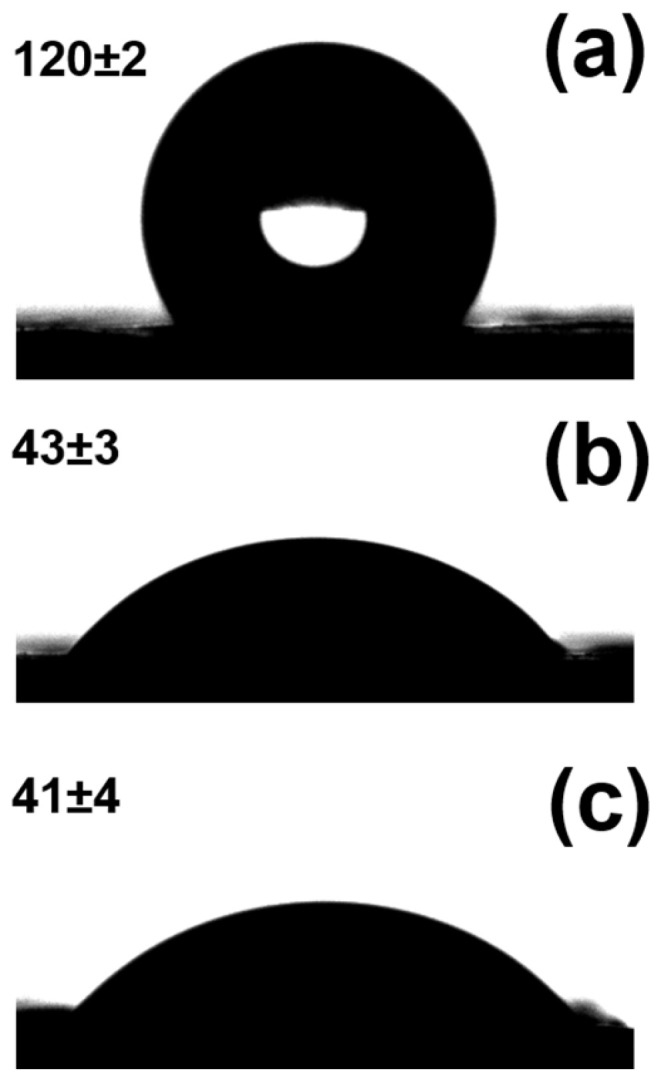
Water contact angle measurements of PCL-ref, (**a**) PCL-COOH (**b**), and PCL-COOH-PRP (**c**).

**Figure 2 ijms-24-08262-f002:**
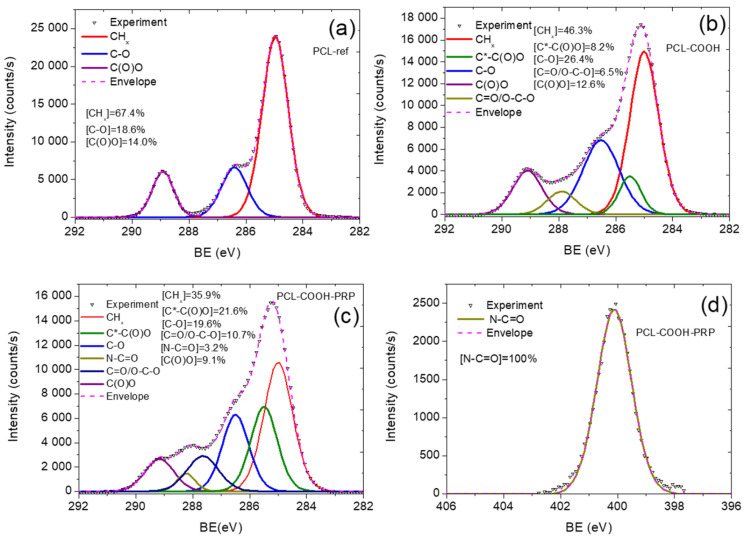
The XPS C1s spectra of as-prepared PCL nanofibers PCL-ref (**a**), nanofibers after deposition of COOH plasma layer PCL-COOH (**b**), and after immobilization of PRP PCL-COOH-PRP (**c**). The XPS N1s spectrum of PCL-COOH-PRP is shown in (**d**).

**Figure 3 ijms-24-08262-f003:**
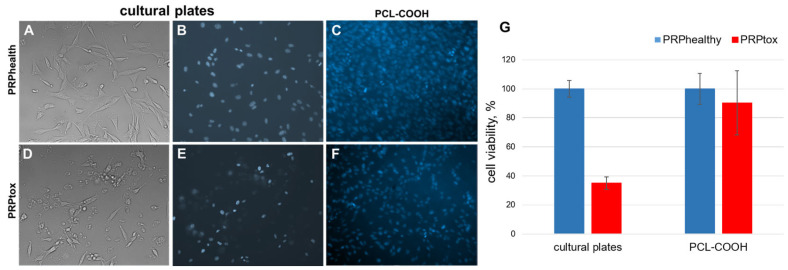
The influence of immobilization of PRPtox on PCL-COOH nanofibers on MSCs viability. (**A**–**F**) Representative photographs of MSCs cultivated on cultural plates with addition of PRP of healthy person (PRPhealth) and PRP T2DM patients displaying toxic effects (PRPtox) and on PCL-COOH-PRPhealthy and PCL-COOH-PRPtox. (**A**,**D**) Morphology of MSCs. (**B**,**C**,**E**,**F**)—The cell nuclei were stained with Hoechst 33342. (**G**) The number of MSCs was defined by counting all fields of view after cell nuclei staining with Hoechst 33342 (the nanofibers had round shape with d = 0.5 cm). The number of cells in control was set 100%. The data are presented as mean ± SD, *n* = 3.

**Figure 4 ijms-24-08262-f004:**
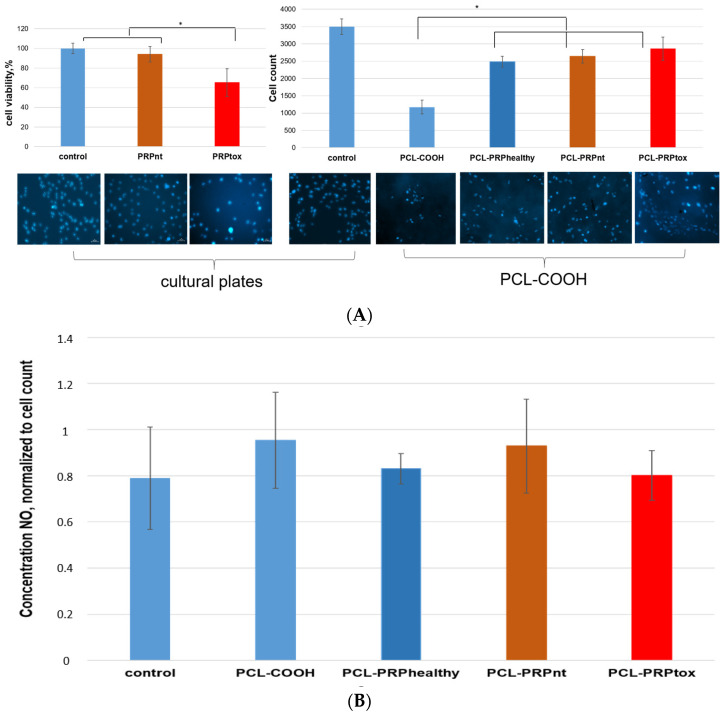
The viability (**A**) and NO secretion (**B**) of endothelial cells growing in cultural media with added of PRP T2DM samples and on PCL-COOH-PRP nanofibers. Cell viability assayed by MTT assay. The number of cells cultivated on PCL-COOH-PRP for 3 days was defined by counting all fields of view after cell staining with Hoechst 33342 (the nanofibers had round shape with d = 0.5 cm). At the same time, the NO concentration in cultural medium was measured using Griess reagent. In the experiment the following groups were used: PCL with covalently attached PRP of healthy donors (PRPhealthy) or PRP of patients with T2DM displaying nontoxic (PRPnt, n = 10) or toxic (PRPtox, n = 10) effects on MSCs according to [27]. As the control group PCL without PRP (PCL-COOH) and cultural plates treated with type IV collagen (control) was used. The data are presented as mean ± SD. *—*p* < 0.01.

**Figure 5 ijms-24-08262-f005:**
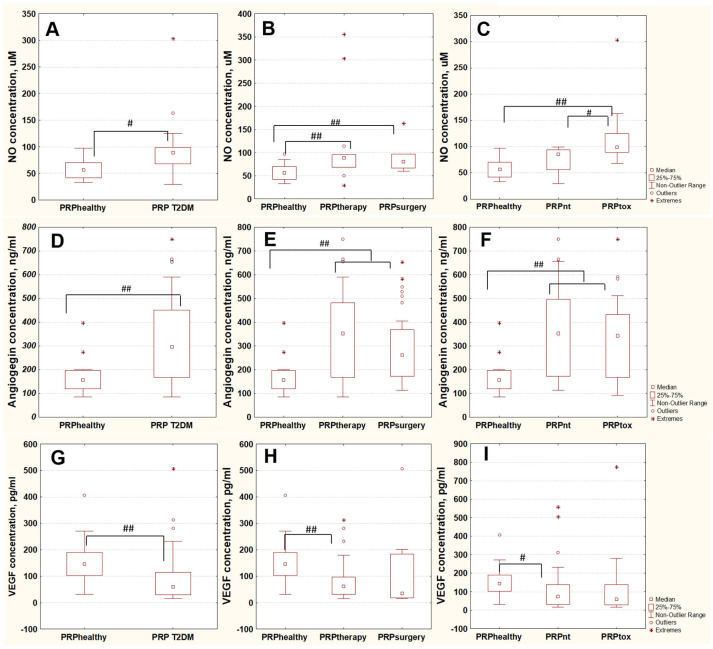
Plasma concentration of NO (**A**–**C**), angiogenin (**D**–**F**), and VEGF (**G**–**I**) in PRP samples. All PRP plasma samples were divided on the following groups: PRP of healthy donors (PRPhealthy, n = 15), PRP of patients with T2DM (PRP T2DM, n = 115); PRP of T2DM patients without (PRPtherapy, n = 60) or with DFU (PRPsurgery, n = 55); PRP of T2DM patients displaying nontoxic (PRPnt, n = 70) or toxic (PRPtox, n = 45) effects on MSCs according to [27]. The data are presented as median ± 25–75% quartiles. #—*p* < 0.05, ##—*p* < 0.01.

**Table 1 ijms-24-08262-t001:** Atomic composition of samples determined by XPS analysis.

Sample	C	O	N	S
PCL-ref	77.3	22.7	0.0	0.0
PCL-COOH	69.3	30.2	0.5	0.0
PCL-COOH-PRP	70.7	17.2	11.8	0.3

## Data Availability

Data are available from corresponding author upon a reasonable request.

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
