# Peer review of "PRP of T2DM Patient Immobilized on PCL Nanofibers Stimulate Endothelial Cells Proliferation"

_ijms, 2023, doi:10.3390/ijms24098262_

Round 1

Reviewer 1 Report

In this study, Anastasiya Solovieva et al. investigated the NO, angiogenin, and VEGF levels in T2DM PRP samples and found the toxicity of T2DM PRP correlates with higher levels of NO in the samples. The authors showed that the immobilization of T2DM PRP on PCL nanofibers has a positive influence on adhesion and proliferation of endothelial cells as compared to control PCL-COOH nanofibers. However, there are only 1 table and 2 figures in the manuscript, which is too preliminary as a full-length article. Therefore, I do not recommend its publication in Int. J. Mol. Sci. in the current form.

My comments:

1.       Methods for WCA and XPS analysis are missing in Materials and Methods.

2.       Page 3, line 113: The sentence “All nitrogen atoms have amide functions.” does not make logical sense.

3.       Page 3, line 121-122: The statement “the increase is determined by the contribution of PRP of toxic group (PRPhealthy/PRPtox, p=0.005).” is difficult to understand.

4.       Please add sample sizes in figure captions.

5.       Page 4, line 134, and page 5, line 170: The sentence “Statistical analysis of data was performed using the STATISTICA 8 program” should be presented in Materials and Methods.

6.       Page 5, line 180-181: The authors demonstrated the addition of 5% T2DM PRP to the culture medium of MSCs led to a decrease in cell viability in 39% of PRP samples in a previous study. However, this does not necessarily mean that PRP samples are toxic to endothelial cells. Therefore, the results presented in Figure 2a did not support the conclusion that “the immobilization of T2DM 180 PRP on PCL nanofibers allows to remove the cytotoxicity of PRP (page 5, line 180-181)”.

7.       Based on results in Figure 1, the authors concluded that the toxicity of T2DM PRP correlates with higher levels of NO in the samples. However, in Discussion (page 6, line 210-225), it was claimed that a decrease in NO secretion is the cause of endothelial dysfunction. The authors are invited to explain this discrepancy.

8.       Page 9, line 341-342: Mann-Whitney U test is only applicable to statistical analyses between two groups. However, there were several analyses containing more than two groups. Please correct the statistical methods where necessary.

9.       Please check the grammar and spelling. For example:

l  Page 1, lines 31: from 371 million in 2013 to 463 millionin 2019 worldwide

l  Page 3, lines 122: twith >>> with.

Author Response

Dear reviewer,

Authors thanks you for the detailed and very helpful review of our manuscript that definitely was very essential to improve the quality of our work. Please find our answers highlighted in yellow. We also added the PDF version for your convenience that included the Figure provided in answers.

 Answer to general comment

The authors showed that the immobilization of T2DM PRP on PCL nanofibers has a positive influence on adhesion and proliferation of endothelial cells as compared to control PCL-COOH nanofibers. However, there are only 1 table and 2 figures in the manuscript, which is too preliminary as a full-length article. Therefore, I do not recommend its publication in Int. J. Mol. Sci. in the current form.

First of all, we would like to thank the reviewer for his great work and his valuable comments. We significantly improved our manuscript, introduced two more multiple figures (WCA and XPS analyses), improved the discussion section.  We would like also to emphasize that we employed only most important data that shows differences, trends and helped us to make the conclusions. We avoided the figures that were not informative. We believe that nowadays all papers tend to be more focused and “crispy” and with minimum figs ad text to help the reader to get the information quickly.

Answeres to focussed comments:

  1. Methods for WCA and XPS analysis are missing in Materials and Methods.

XPS and WCA are now provided in the materials and methods.

  1. Page 3, line 113: The sentence “All nitrogen atoms have amide functions.” does not make logical sense.

We have added the N1s curve fitting and deleted this sentence.

  1. Page 3, line 121-122: The statement “the increase is determined by the contribution of PRP of toxic group (PRPhealthy/PRPtox, p=0.005).” is difficult to understand.

We corrected the sentences: The data showed (Fig. 3 A, C) that the level of NO in T2DM PRP is reliably higher (p=0.003) as compared to PRP of healthy donors, and the increase of NO concentration is determined by the contribution of PRP of toxic group in T2DM PRP group

  1. 4.       Please add sample sizes in figure captions.

Correction was done in the revised version

  1. Page 4, line 134, and page 5, line 170: “The sentence “Statistical analysis of data was performed using the STATISTICA 8 program” should be presented in Materials and Methods. in

  Correction was done in the revised version:  delete “The sentence “Statistical analysis of data was performed using the STATISTICA 8 program” in    Page 4, line 134, and page 5, line 170.  The sentence “Statistical analysis of data was performed using the STATISTICA 8 program” takes place in Materials and Methods (line 341). Add subsection 4.7 Statistical Analysis

  1. Page 5, line 180-181: The authors demonstrated the addition of 5% T2DM PRP to the culture medium of MSCs led to a decrease in cell viability in 39% of PRP samples in a previous study. However, this does not necessarily mean that PRP samples are toxic to endothelial cells. Therefore, the results presented in Figure 2a did not support the conclusion that “the immobilization of T2DM 180 PRP on PCL nanofibers allows to remove the cytotoxicity of PRP (page 5, line 180-181)”.

PRPtox caused the death of up to 60% of cells, moreover, the cells, according to morphological features, died from necrosis (fig1). We studied the viability of MSCs seeded on PCL-COOH-PRPtox. Our data showed that after immobilization of PRP proteins on PCL nanofibers their toxic effects is removed. It is unlikely that this effect is cell-line specific. Endothelial cells more sensitive and exacting  of cultivation conditions, so we believe that we have reason to believe that the toxicity will be similar on endothelial cells.

Please See the attached PDF file.

Figure 1. Cytotoxic effect of PRPtox on MSCs. (see the PDF attached)

  1. Based on results in Figure 1, the authors concluded that the toxicity of T2DM PRP correlates with higher levels of NO in the samples. However, in Discussion (page 6, line 210-225), it was claimed that a decrease in NO secretion is the cause of endothelial dysfunction. The authors are invited to explain this discrepancy.

The text in Discussion was corrected:

The level of NO rises, while the level of reactive oxygen species (ROS) rises as well. ROS oxidize NO, forming peroxynitrites, which in turn oxidize the zinc thiolate center of endothelial NO synthases (eNOS) [35] . Dysfunctional eNOS leads to a decrease in the secretion of NO by endothelial cells and its biological activity [20] . According to our data, the NO concentration in PRP T2DM (Figure 3) is more than 10-fold higher than that secreted by endothelial cells (Figure 4b). Our data confirm that the main source of NO is not endothelial cells but inflammatory cells, in particular macrophages.

  1. Page 9, line 341-342: Mann-Whitney U test is only applicable to statistical analyses between two groups. However, there were several analyses containing more than two groups. Please correct the statistical methods where necessary.

The statistical methods was corrected

  1. Please check the grammar and spelling. For example:

l  Page 1, lines 31: from 371 million in 2013 to 463 millionin 2019 worldwide  in      Done:  add space between million and in

l  Page 3, lines 122: twith >>> with. 

 Done: delete t

Reviewer 2 Report

In this manuscript, Anastasiya Solovieva and coworkers have developed polycaprolactone (PCL) nanofibers modified with COOH groups to covalently immobilize platelet-rich plasma (PRP) from Type2 Diabetes Mellitus (T2DM) patients. The nanofibers were evaluated for the viability and proliferative activity of endothelial cells. The authors also investigated the content of nitric oxide, and angiogenic factors in the blood samples of T2DM patients. The authors have demonstrated that these modified PCL nanofibers could potentially be used as a dressing for Diabetic Foot Ulcers (DFU), a complication of T2DM. Overall, this manuscript is well-written and it can be considered for publication once the following minor concerns are addressed- 

  1. Line 57: Please complete the sentence.

  1. Line 96: Please complete the sentence.

Author Response

Dear reviewer!

Authors thanks you for the detailed and very helpful review of our manuscript that definitely was very essential to improve the quality of our work.

  Line 57: Please complete the sentence. Done: add dysfunction

  1. Line 96: Please complete the sentence. Done: add study

Reviewer 3 Report

In the manuscript, the authors immobilized the platelet-rich plasma (PRP) proteins from Type2 Diabetes Mellitus (T2DM) on polycaprolactone (PCL) nanofibers to remove the toxic metabolites for finding a perspective approach for diabetic foot ulcers (DFU) treatment. The authors revealed that PCL-PRP nanofibers use as matrix enhanced the adhesion and proliferation of endothelial cells independently on PRP origin. Therefore, the manuscript addresses the potential interest of the study.

However, the provided data sets to support the conclusion is not sufficient and thus it seems premature to proceed with the manuscript based on the current results. Hence, I recommend revision of manuscript to refine.

1.       The authors should add the contact angle data in the manuscript for the control as well as modified PCL-fibers.

2.       Plasma proteins are very large in size. The authors used only 15 mins for covalent coupling for PCL-COOH with plasma protein. What is the percentage coupling in the final product? Is there any PCL-COOH left uncoupled? The authors should submit the characterization NMR and FT-IR showing all of PCL-COOH was consumed in the coupling reactions.

3.       PCL-COOH-PRP should be PCL-CONH-PRP as during the coupling reaction -COOH group with change into -CONH-

4.       Sulphur was detected in XPS of PCL-COOH-PRP. R-SH groups are well known to couple with R-COOH to from thioester. Is this happening during coupling? The authors should comment on it.

5.       What is the concentration of cells on culture plates? The authors should add the data for endothelial cells grow on cultural plates treated with type IV collagen for better comparison and understanding.

Author Response

Dear Reviewer,

Thank you so much for your very valuable comments that were essential to improving our work. We added two figures and improved our Results and Discussion section, and we kept track of all our changes. Please see our answers below.

  1. The authors should add the contact angle data in the manuscript for the control as well as modified PCL-fibers.

The values of WCA were provided in Figure 1 in the revised version

  1. Plasma proteins are very large in size. The authors used only 15 mins for covalent coupling for PCL-COOH with plasma protein. What is the percentage coupling in the final product? Is there any PCL-COOH left uncoupled? The authors should submit the characterization NMR and FT-IR showing all of PCL-COOH was consumed in the coupling reactions.

Although the proteins are of relatively large size, the reaction takes place very quickly. We have study this process using multiple methods including FT-IR, XPS, EDX, special curve fitting for XPS, etc. In our very recent paper we have performed the C1s curve fitting using special model and we have found that the surface is covered by proteins with more than 60% of coverage. Moreover, our layers exhibited very high coverage even after 7 days of soaking of our samples in PBS. Therefore, the 15 minutes immobilization procedure provides very efficient immobilization. Please see the reference.

https://www.mdpi.com/2073-4360/15/6/1440

However, this methodology is based surface science data and we believe that for the IJMS audience and for this particular paper those details are not very relevant. However, a keen reader can always explore these results in the mentioned reference.

Additionally, we believe that NMR and FT-IR are less appropriate to characterize the surface chemistry as they are more appropriate for study of bulk materials. Moreover, the FT-IR has superposition of C=O coming from COOH and COOR groups (from PCL). Therefore, the features of our COOH layers are less visible as compared with PCL peaks from nanofibers. As for the NMR, the solid-state NMR will be also less informative as it will collect the signal from the PCL, rather our thin plasma coating.

  1. PCL-COOH-PRP should be PCL-CONH-PRP as during the coupling reaction -COOH group with change into -CONH-

Indeed, we agree with the reviewer that the chemistry after immobilization will change from COOH to CONH. However, the symbols in the assignment very rather attributed to the following of the stages of each process, than the composition of the samples. In other words, the PCL stands for the PCL nanofibers, COOH stands for the deposition of COOH plasma layer, PRP for the PRP immobilization.

  1. Sulphur was detected in XPS of PCL-COOH-PRP. R-SH groups are well known to couple with R-COOH to from thioester. Is this happening during coupling? The authors should comment on it.

The presence of the sulfur is related to the sulfur-containing aminoacids (cysteine, etc.) in the proteins structure. We have not employed any SH chemistry in our process.

  1. What is the concentration of cells on culture plates? The authors should add the data for endothelial cells grow on cultural plates treated with type IV collagen for better comparison and understanding.

The  concentration of cells on culture plates were provided in Figure 4 in the revised version

Reviewer 4 Report

Overall, this research article will help to address the wound healing in Type 2 Diabetes Mellitus patients. Authors conducted study in human patients and cell tests, which reflects the practical significance of the results. This article fits the aim & scope of the journal IJMS.

Comments:

Line 31: there should be space between million and in

Line 85: there should be space between the and cytotoxicity

Line 96: should be in our previous study

Line 122:  twith?

Line 125 and 126: is not eligible? is it not negligible or not eligible?

Line 167: healthy

Figure 1: Figure caption mentioned from (A-I) but the figure represents i, should be consistent, change to I will help readers.

Figure 1: A, C  authors mentioned PRP health whereas in B PRP healthy. Authors should use either one of them consistently, so it won't cause confusion. 

Figure 1: Y axis font size (concentration) should be consistent (for D one size and others different). Be consistent.

Author Response

Dear Reviewer,

Thank you so much for your very valuable comments that were essential to improving our work. We added two figures and improved our Results and Discussion section, and we kept track of all our changes. Please see our answers below.

Comments:

Line 31: there should be space between million and in      Done:  add space between million and in

Line 85: there should be space between the and cytotoxicity done: add space between the and cytotoxicity

Line 96: should be in our previous study  done: add study

Line 122:  twith?  Done: delete t

Line 125 and 126: is not eligible? is it not negligible or not eligible? Done correct is not eligible

 donLine 167: healthy done add t

Figure 1: Figure caption mentioned from (A-I) but the figure represents i, should be consistent, change to I will help readers.

Correction was done in the revised version

Figure 1: A, C  authors mentioned PRP health whereas in B PRP healthy. Authors should use either one of them consistently, so it won't cause confusion. 

Correction was done in the revised version

Figure 1: Y axis font size (concentration) should be consistent (for D one size and others different). Be consistent.

Correction was done in the revised version. Please keep in mind that Figure 1 now is Figure 3.

Round 2

Reviewer 1 Report

The authors have made significant efforts to address my comments. However, further revisions are required before the manuscript can be considered for publication.

My comments:

1.       In response to my comment #6, the authors claimed they studied the viability of MSCs seeded on PCL-COOH-PRPtox. Their data showed after immobilization of PRP proteins on PCL nanofibers their toxic effects is removed. They speculated the effect is not cell-line specific and believe the toxicity will be similar on endothelial cells. What I want to say is that if the authors think the effect is not cell-line specific, the conclusion “the immobilization of T2DM 180 PRP on PCL nanofibers allows to remove the cytotoxicity of PRP” should be considered as a verification of their previous study with MSCs, but not a new finding of this study. Meanwhile, the data in Figure 4a would become needless. Personally, I still suggest the authors provide direct evidence to demonstrate PRP samples are toxic to endothelial cells.

2.       Page 10, line 482: In this study, there are both statistical analyses containing two groups and statistical analyses containing more than two groups. The Kruskal-Wallis test is only applicable to the latter. Moreover, the authors should explain why non-parametric test is used.

Author Response

Dear reviewer!

Authors thanks you for the detailed and very helpful review of our manuscript that definitely was very essential to improve the quality of our work.

 Comment 1.

In response to my comment #6, the authors claimed they studied the viability of MSCs seeded on PCL-COOH-PRPtox. Their data showed after immobilization of PRP proteins on PCL nanofibers their toxic effects is removed. They speculated the effect is not cell-line specific and believe the toxicity will be similar on endothelial cells. What I want to say is that if the authors think the effect is not cell-line specific, the conclusion “the immobilization of T2DM 180 PRP on PCL nanofibers allows to remove the cytotoxicity of PRP” should be considered as a verification of their previous study with MSCs, but not a new finding of this study. Meanwhile, the data in Figure 4a would become needless. Personally, I still suggest the authors provide direct evidence to demonstrate PRP samples are toxic to endothelial cells.

ANSWER:

We do agree with the reviewer’s comment. To confirm the presented conclusion “the immobilization of T2DM PRP on PCL nanofibers allows to remove the cytotoxicity of PRP”, we presented in the article new findings obtained with the MSCs. Accordingly, the conclusion on reducing the toxicity of PRP when immobilizing on nanofiber is made based on these results. We introduced new paragraphs on page 5. All new changes are highlighted in the revised manuscript.

For yours convenience the revised paragraph with newly introduced Fig3 is reported below.

Here we compared the toxic effects of PRPtox versus PRPhealthy samples, which were added to cultural medium or immobilized on PCL-COOH on MSCs (Figure 3).  

Figure 3. The influence of immobilization of PRPtox on PCL-COOH nanofibers on MSCs viability. A-F Representative photographs of MSCs cultivated on cultural plates with addition of PRP of healthy person (PRPhealth) and PRP T2DM patients displaying toxic effects (PRPtox) and on PCL-COOH-PRPhealthy and PCL-COOH-PRPtox. A, D Morphology of MSCs. B, C, E, F –The cell nuclei were stained with Hoechst 33342. G. The number of MSCs was defined by counting all fields of view after cell nuclei staining with Hoechst 33342 (the nanofibers had round shape with d=0.5 сm). The number of cells in control was set 100%. The data are presented as mean ± SD, n=3.

The data showed that the cultivation of MSCs in medium with PRPtox led to cell death by necrosis (Figure 3 D). After immobilization of PRPtox on PCL-COOH and following cultivation of MSCs on it, the nuclei morphology corresponds to the morphology of viable cells (structure, shape, density, the presence of mitosis). The number of living cells was evaluated by counting the nuclei of living cells using fluorescent microscopy. The data in Figure 3 G showed a significant increase in cell number growing on PCL-COOH-PRPtox versus on cultural plates with PRPtox addition. Our data suggest that immobilization of PRPtox on PCL-COOH eliminates the toxic effects and, at the same time, maintains the proliferative activity of MSCs.

Since activation of angiogenesis is necessary for DFU healing, endothelial cells were used as a model cell system. We studied the influence of all pooled PRP samples immobilized on nanofibers on the viability and functional activity of the endothelial cells.

The following groups were defined: PRPnt, nontoxic (according to [26] cell viability is more than 60%), PRPtox, toxic group (cell viability is less than 60%), or PRPther, therapeutic (from patients without DFU), and PRPsur, surgery group (from patients with DFU). PRP from healthy donors (PRPhealthy) was used as a control group. A group of PRP patients with T2DM was designated as PRP T2DM. The levels of NO metabolites and VEGF, angiogenin growth factors, were measured.

Comment 2

Page 10, line 482: In this study, there are both statistical analyses containing two groups and statistical analyses containing more than two groups. The Kruskal-Wallis test is only applicable to the latter. Moreover, the authors should explain why non-parametric test is used.

ANSWER:

The correction was done in «Materials and Methods» in the revised version.

The Shapiro-Wilk test was used to control the normal distribution of all variables. Since the obtained data did not show a normal distribution, a nonparametric test was appropriate, namely the non-parametric Mann-Whitney U test was used for comparing two groups and Kruskal-Wallis test was used for comparing more than two groups.”

 Our changes are highlighted in yellow.

Reviewer 3 Report

I recommend this manuscript for the publication.

Author Response

We are grateful to the Reviewer for his great remarks and suggestions provide during peer-review. All questions were addressed and all changes in the manuscript are highlighted.

Round 3

Reviewer 1 Report

Figure 3, which demonstrates immobilization of PRPtox on PCL-COOH eliminates toxic effects of PRPtox to MSCs, is added in the revised manuscript. While this partially addresses my concern (comment #1) in the last review report. I would like to emphasize that it is still not direct evidence. “PRPtox is toxic to endothelial cells” is an important precondition the study relies on, as the aim of it is to investigate the effects of PRP immobilization on endothelial cells. Using another cell type to demonstrate toxic effects of PRPtox (to endothelial cells) will compromise the rigor of the manuscript. Hence, I strongly suggest the authors provide direct evidence regarding the toxic effects of PRPtox to endothelial cells.

Author Response

Dear reviewer,

We are grateful for your great attention and significant contribution that ware very essential to improve our manuscript. In accordance with your recommendation, we conducted additional experiments to determine the toxicity of the PRP-tox samples on endothelial cells. Please see the newly added Figure 4a. The relevant information has been included in the article and all our changes were tracked. The following was added to our manuscript:

“We studied the cytotoxicity of PRPtox and PRPnt samples using endothelial cell lines. The cytotoxicity was evaluated on cultural plates and also using PCL-COOH samples with immobilized PRP, and the influence of these samples on cell viability was investigated. The effect of PRP samples on cell viability was determined by the MTT test and by counting all fields of view after cell staining with Hoechst 33342 fluorescent dye. It was shown that PRPtox led to a significant decrease in the proliferative and mitochondrial activity of endothelial cells without signs of cell death by necrosis (Figure 4a).”